# Randomized routing of virtual machines in IaaS data centers

Hadi Khani[1] and Hamed Khanmirza[2]

[1] Department of Engineering, Islamic Azad University Garmsar Branch, Garmsar, Semnan, Iran
[2] Department of Computer Engineering, K. N. Toosi University of Technology, Tehran, Tehran, Iran



## ABSTRACT

Cloud computing technology has been a game changer in recent years. Cloud computing providers promise cost-effective and on-demand resource computing for their users. Cloud computing providers are running the workloads of users as virtual machines (VMs) in a large-scale data center consisting a few thousands physical servers. Cloud data centers face highly dynamic workloads varying over time and many short tasks that demand quick resource management decisions. These data centers are large scale and the behavior of workload is unpredictable. The incoming VM must be assigned onto the proper physical machine (PM) in order to keep a balance between power consumption and quality of service. The scale and agility of cloud computing data centers are unprecedented so the previous approaches are fruitless. We suggest an analytical model for cloud computing data centers when the number of PMs in the data center is large. In particular, we focus on the assignment of VM onto PMs regardless of their current load. For exponential VM arrival with general distribution sojourn time, the mean power consumption is calculated. Then, we show the minimum power consumption under quality of service constraint will be achieved with randomize assignment of incoming VMs onto PMs. Extensive simulation supports the validity of our analytical model.

## INTRODUCTION

Infrastructure-as-a-Service (IaaS) cloud providers (CPs), such as Amazon, Google and Microsoft, have huge data centers to provide on demand virtual machines (VMs) to their customers. An important issue for such data centers is to determine the server to which an incoming VM should be placed in order to optimize a given performance criterion. The CP has a variety of challenges, such as higher resource utilization, less cooling expenses and lower operation expenses. Fortunately, all of these efficiency metrics are positively correlated. Less power consumption means less operational expense, less cooling bills and higher utilization in the data center. This lets us choose the power consumption as the key metric representing others. On the other hand, cloud users who run their applications on VMs have their own concerns with quality of service. The resource management of CP has the chance to revise the initial placement of VMs onto PMs by live migrating techniques or dynamic consolidation. Considering live migration, the problem of VM placement can be divided in two parts as pictured in Fig. 1.

Corresponding author
Hadi Khani, hkhani@ut.ac.ir

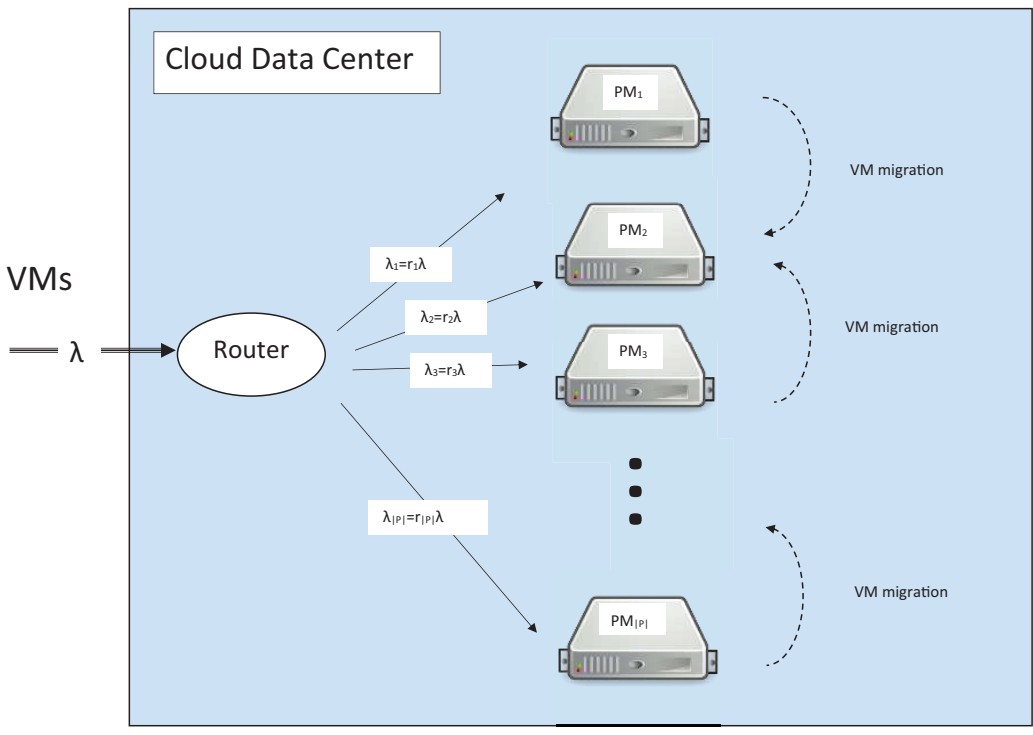

**Figure 1 Randomized router.**

1. The routing of arriving VMs onto PMs

2. The optimization of the current VM placement by VM migration

According to this scenario, a VM request transmitted by a user to the data center is routed to the proper PM in the first step and then its placement can be optimized later.

The optimization of the current VM placement in the data center is analogous to the NP-hard "Bin Packing Problem." In this problem, a given set of items of variable size should assigned to the minimum number of bins taken from a given set. The VMs experience dynamic workloads, which means that the resource usage by a VM arbitrarily varies over time. In fact, the data center resource manager does not have the complete knowledge of the future resource usage (size) of VMs. The placement of VMs is monitored continuously and is tuned through the migration procedure. The virtualization technology let VM migrates (moves) between PMs on the fly. The migration of VM can be advantageous either when the resources utilization is too low, meaning that the PM is highly underutilized, or when it is too high, possibly causing overload situations and service level agreement violations (SLAVs). The optimization problem of VM placement problem is so complex that centralized and deterministic solutions are practically useless in large data centers with hundreds or thousands of servers as shown in several researches like (*Wang & Gelenbe, 2015*; *Shojafar, Cordeschi & Baccarelli, 2016*; *Wang, Jiang & Wu, 2016*). The centralized and deterministic algorithms may be appropriate in data centers with a limited number of servers, but may become inefficient in large and

very large data centers, due to the complexity of the problem and the need for the simultaneous migrations of a large number of VMs.

These decentralized approaches have some side effect on routing procedure in the first part. The router does not have complete knowledge of the current placement of cloud data center. It is noteworthy that the router neither does not know about the size, the exact arriving time and sojourn time of the future VMs. These facts justify the stochastic modeling and analyzing of the router performance.

In this paper, we focus on the first problem: the problem of routing arriving VM to the host. We calculate probability of SLAV as well as total power consumption in a cloud data center using tools of queueing theory. A cloud data center differs from traditional queueing systems. First, a cloud data can have a large number of PMs; traditional queueing analysis rarely consider system of this size. Second, VM sojourn time must be modeled by general distribution instead of convenient exponential distribution. These differences pose significant challenges to the analysis. We use a novel approach to respond to these challenges. Our contributions in this paper are:

1. We model the cloud data centers as a group of $M/G/n/n$ queuing systems with single task arrivals and a task buffer of finite capacity;
2. We define a novel optimization problem to minimize the power consumption under an explicit QoS goal for any VM consolidation system;
3. We find the optimal routing policy using numerical methods.

Analytical results are validated through discrete event simulation. Then, we compare our result with some benchmark algorithm for Google workload. The remainder of the paper is organized as follows. The "Related Work" section gives an overview of existing work on cloud performance evaluation and performance characterization of $M/G/n/n + r$ queuing systems. It also introduces some heuristic algorithms for VM consolidation that we use for comparison. In the "System Model" section we discuss our analytical model in detail. We solve our optimization problem in order to obtain desired performance metrics in the "Optimization Problem" section. In the "Simulation Results" section, we present and discuss analytical as well as simulation results. We conclude the paper with the section "Conclusion" discussing the results and future research directions.

## RELATED WORK

Prior approaches to VM placement in the literature can be broadly divided into two categories: rigorous analytical approach and heuristic algorithms. One of the first works on analysis of performance issues in VM placement has been performed by *Yang et al. (2009)*. They obtained the distribution of response time for a cloud data center modeled as an $M/M/n/n + r$ queueing system. They assumed both interarrival and service times are exponentially distributed, and the system has finite buffer of size $n + r$. The response time was broken down into waiting, service and execution periods, assuming that all three periods are independent, which is unrealistic.

By relaxing the assumption that the service times are not exponential, one can construct an accurate and close-to-reality model at the expense of greater complexity in the analysis. Most theoretical analyses have relied on extensive research in performance evaluation of $M/G/n$ queuing systems (*Ma & Mark, 1995*; *Miyazawa, 1986*; *Yao, 1985*). However, the probability distributions of response time and queue length in $M/G/n$ and $M/G/n/n + r$ cannot be obtained in closed form, which necessitates the search for a suitable approximation. An approximate solution for steady-state queue length distribution in an $M/G/n$ system with finite waiting space is described in *Kimura (1996)*. The proposed approach is exact for $M/G/n/n + r$ when $r = 0$.

A similar approach for $M/G/m$ queues is proposed in *Kimura (1983)*. In this work, analysis is extended to compute the blocking probability and, thus, determines the smallest buffer capacity such that the rate of lost tasks remains below a predefined level. In *Nozaki & Ross (1978)*, another approximation for the average queuing delay in an $M/G/n/n + r$ queue was proposed. The approximation is based on the relationship of joint distribution of remaining service time to the equilibrium service distribution. Another approximation for the blocking probability is based on the exact solution for finite capacity $M/M/n/n + r$ queues (*Smith, 2003*). Again, the estimate of the blocking probability is used to guide the allocation of buffers so that the blocking probability remains below a specific threshold.

Most of above findings rely on some approximations. Approximations are reasonably accurate only when the number of servers is comparatively small, typically below 10 or so. In addition, approximation errors are high when the traffic intensity is small as stated in *Boxma, Cohen & Huffels (1979)*, *Kimura (1983)*, and *Tijms, Van Hoorn & Federgruen (1981)*. As a result, we cannot apply the above results directly for performance analysis of CP data center when one or more of the following holds: the number of servers is very large or the distribution of service times is unknown and does not follow any of the "well-behaved" probability distributions such as exponential distribution.

As we use the $M/G/n/n$ queueing system to model a physical machine (PM) and not the whole data center, our analysis is suitable for performance analysis of cloud scale data centers. In addition, we study $M/G/n/n$ in steady station setting. *Kimura (1996)* has proposed an exact closed form for queue length distribution in an $M/G/n/n$. In this paper, we use this closed form in defining optimization problem (*Kimura, 1996*) which let us apply numerical computation for analyzing the performance of the whole data center in next step.

In order to compare the performance of randomized router in practice, we have chosen two algorithms from heuristic algorithms as benchmark: power aware best fit decreasing (PABFD) (*Beloglazov & Buyya, 2012*) and modified throttled (MT) (*Domanal & Reddy, 2013*).

As mentioned before, the VM placement can be seen as a bin packing problem with variable bin sizes and prices, where bins represent the PMs; items are the VMs that have to be allocated; bin sizes are the available CPU capacities of the PMs; and prices correspond to the power consumption by the nodes. As the bin packing problem is NP-hard, to solve it *Beloglazov & Buyya (2012)* apply a modification of the best fit decreasing algorithm

that is shown to use no more than $\frac{11}{9}$ OPT $+\,1$ bins (where OPT is the number of bins provided by the optimal solution) (*Yue, 1991*). In PABFD, they sort all the VMs in the decreasing order of their current CPU utilizations and allocate each VM to a host that provides the least increase of the power consumption caused by the allocation. In each round of PABFD all VMs are placed again. The number of VM migrations skyrockets in PABFD and it is not practical in a real large-scale data center.

Modified throttled algorithm maintains an index table of PMs and also the state of PMs (*Domanal & Reddy, 2013*). There has been an attempt made to improve the response time and achieve efficient usage of available PMs. Proposed algorithm employs a method for selecting a machine for hosting arriving VM of user where, machine at first index is initially selected depending upon the state of the machine. If the machine is available, it is assigned with the request and id of machine is returned to data center resource manager, else −1 is returned. When the next request arrives, the machine at the index next to already assigned machine is chosen depending on the state of machine and follows the above step. This method needs to keep an updated index table of state of machines. In large data centers this task is not trivial, in particular when you taking into account the decentralized consolidation of VMs. It is important that both MT and PABFD is not practical in real scenario, and here we use them just as idealistic benchmark algorithms.

## SYSTEM MODEL

Consider a IaaS data center consisting of a set of $|P|$ PMs. Let $P = \{1,2,...,|P|\}$ denote index set of the set of PMs. Users request VMs to the provider. New request is either admitted or rejected in the admission control phase. An admitted request moves to the placement phase, where it will be assigned to one of PMs (*Carvalho, Menasce & Brasileiro, 2015*). We suggest a randomized router after the admission. The router sends incoming VMs to PM $i$ with probability $r_i$, for all $i \in P$. The vector $\vec{r} = \{r_1, r_2, \ldots, r_{|P|}\}$ is a probability vector and satisfies $\sum_{i \in P} r_i = 1$. VMs have independent and identically distributed service (sojourn) time with mean $1/\mu$. In addition, these sojourn times are independent of the host load. Assume that the utilization demand of all VMs is equal to one. Extensive analysis of huge data centers shows that majority of the VMs have approximately the same utilization (*Reiss, Wilkes & Hellerstein, 2011*). This observation supports our assumption. Assume that each PM only hosts at most $n$ VMs. In the case a VM is assigned to an already full PM, the PM is overloaded. An overloaded PM degrades the QoS of the end user and we can assume this event as an SLAV (*Domanal & Reddy, 2013*; *Beloglazov & Buyya, 2012*). It should be remembered that the router is after admission control and admitted VMs could not be queued. All VMs arrive at the data center according to Poisson process with rate λ, thereby VMs arrive at PM $i$ according to Poisson process with rate $\lambda_i = \lambda\, r_i$, for all $i \in P$ (*Wang, Chang & Liu, 2015*). These processes are independent (see Section 6.4 in *Trivedi (2001)*). The whole data center can be modeled as a group of independent $M/G/n/n$ (also known as generalized Erlang loss system) systems that work in parallel.

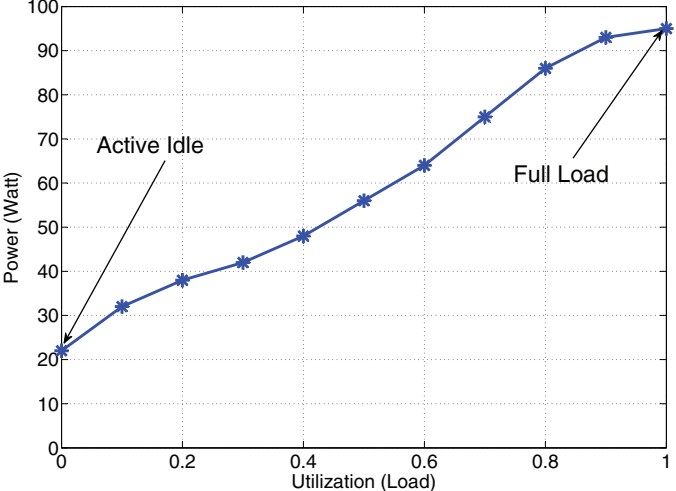

**Figure 2  Linear power consumption.**

## Generalized Erlang loss system

It is well known that for an *M/G/n/n* system, the steady-state distribution exists and is given by the "product form" as below (*Kelly, 1991*; *Kimura, 1983*).

$$\pi_k = \begin{cases} \pi_0 \left(\dfrac{\lambda}{\mu}\right)^k \dfrac{1}{k!}, & \text{for } k \leq n \\ 0, & \text{for } k > n \end{cases} \tag{1}$$

where $\pi_k$ denotes the steady-state probability that there are $k$ VMs in an *M/G/n/n* system, that is $\pi_k = \lim_{t \to \infty} \pi_k(t)$, and the steady-state probability that *M/G/n/n* is empty ($\pi_0$) is given by

$$\pi_0 = \left[\sum_{k=0}^{n} \left(\frac{\lambda}{\mu}\right)^k \frac{1}{k!}\right]^{-1} \tag{2}$$

Specifically, $\pi_n$ describes the fraction of time that the PM is fully utilized. We call this probability SLAV and it is given by

$$\pi_n = \frac{\dfrac{(\lambda/\mu)^n}{n!}}{\displaystyle\sum_{k=0}^{n} \frac{(\lambda/\mu)^k}{k!}} \tag{3}$$

## Power consumption

We are interested in minimizing the total power consumption of the data center. According to the results of the standard experiments stated in (*spec.org, 2018*) (Fig. 2), the instantaneous power consumption of PM $i$ is a function of utilization level of that PM ($k$) as below

$$W_i = \begin{cases} a + bk, & \text{for } k > 0 \\ 0, & \text{for } k = 0 \end{cases} \tag{4} \\ \tag{5}$$

where $0 \leq k \leq 100$ is integer variable and both $a$ and $b$ are fixed values. In our model, $k$ can be seen as the number of VMs in PM $i$. Then the expected steady state of power consumption will be

$$\mathrm{Exp}[W_i] = \sum_{k=1}^{100} W_{i,k} \pi_{i,k} \tag{6}$$

$$= a(1 - \pi_{i,0}) + b \sum_{k=1}^{100} k \pi_{i,k} \tag{7}$$

where $\pi_{i,k}$ denotes the steady-state probability that the utilization of PM $i$ is $k$ and can be calculated by Eq. (1) for $n = 100$. Note that $\mathrm{Exp}[W_i] = 0$ if and only if $\lambda_i = 0$ ($\pi_{i,0} = 1$). Our objective is to determine the vector $\vec{r}$ that minimizes the total expected steady-state power consumption of the data center, that is,

$$\min_{\vec{r}} \sum_{i \in P} \mathrm{Exp}[W_i] \tag{8}$$

## SLA constraint

We are interested in keeping the probability of SLAV below a given value $\varepsilon$. SLAV happens when a PM $i$ does not have sufficient capacity for a new arriving VM (when M/G/100/100 is in state 100). The SLAV constraint is

$$\mathrm{Pr(SLAV)} = \pi_{i,100} \leq \varepsilon \qquad \forall i \in P \tag{9}$$

where $\pi_{i,100}$ denotes the steady-state blocking probability for PM $i$.

## OPTIMIZATION PROBLEM

In this section, we consider the optimization problem, in which the router decides where each incoming VM will be sent, so as to minimize the total expected power consumption subject to the SLA constraints. The optimization problem can be formulated as follows:

$$\min_{\vec{r}} \quad \sum_{i \in P} \mathrm{Exp}[W_i] \tag{10}$$

$$\mathrm{s.t.} \quad \sum_{i \in P} r_i = 1 \tag{11}$$

$$\pi_{i,100} \leq \varepsilon \; \forall i \in P \tag{12}$$

$$\lambda_i = \lambda r_i \quad \forall i \in P \tag{13}$$

Let us rewrite the optimization problem by changing our *optimization variable* from $\vec{r} = \{r_1, r_2, ..., r_{|P|}\}$ to $\vec{x} = \{x_1, x_2, ..., x_{|P|}\}$ which is defined below

$$x_i = \frac{\lambda_i}{\mu} = \frac{\lambda r_i}{\mu} \tag{14}$$

Using Eq. (14) and putting Eq. (1) for $k = 100$ in Eq. (12) we get

$$\frac{x_i^{100}}{100! \sum_{k=0}^{100} \frac{x_i^k}{k!}} \leq \varepsilon \tag{15}$$

**Table 1 Numerical results for maximum incoming rate ($x_i < \beta$) based on acceptable SLAV probability ($\varepsilon$).**

| SLAV probability ($\varepsilon$) | Maximum rate ($\beta$) |
|---|---|
| 0.1 | 100 |
| 0.05 | 95 |
| $10^{-3}$ | 80 |
| $10^{-4}$ | 73 |
| $10^{-5}$ | 69 |

If we can solve this inequality constraint for $x_i$ then we have a simple inequality constraint in the form of $x_i < f(\varepsilon)$. Fortunately, numerical methods can be used to solve it. We show numerical results $\beta$ for some practical values of $\varepsilon$ in Table 1. For example, in the first row, we have $\beta = 100$ for $\varepsilon = 0.1$. It means that if we send VMs with a rate below 100 ($x_i < \beta = 100$) to PM $i$ then we will guarantee that the probability of SLAV is below 0.1 ($\Pr(\text{SLAV}) < \varepsilon = 0.1$) for that PM. The equivalent optimization problem will be

$$\min_{\vec{r}} \quad f_o(\vec{x}) = \sum_{i \in P} \gamma(x_i) \tag{16}$$

$$\text{s.t.} \quad \sum_{i \in P} x_i = \frac{\lambda}{\mu} \tag{17}$$

$$x_i \leq \beta \quad \forall i \in P \tag{18}$$

in which

$$\gamma(x) = a + \frac{1}{\sum_{k=0}^{100} \frac{x^k}{k!}} \left( -a + b \sum_{k=1}^{100} k \frac{x^k}{k!} \right) \tag{19}$$

The $\gamma(x)$ can be obtained using Eqs. (1), (2) and (7) with ease. We set $a = 22$ and $b = 0.73$ for a PM equipped with Intel Xeon E3 processor (*spec.org, 2018*). Then, we can show that the first order derivative of $\gamma(x)$ is positive ($\gamma'(x) > 0$) and the second order derivative of $\gamma(x)$ is negative ($\gamma''(x) < 0$) for ($0 \leq x \leq 100$) with numerical methods.

**Theorem 1.** *For any x and y ($0 < y \leq x \leq \beta$), there exists $\delta$ ($0 < \delta \leq y$ and $0 < \delta \leq \beta - x$), so that*

$$\gamma(x + \delta) + \gamma(y - \delta) \leq \gamma(x) + \gamma(y) \tag{20}$$

**Proof:** $\gamma''(x) < 0$ means that the derivative ($\gamma'(x)$) is nonincreasing. The condition $y \leq x$ implies that the $\gamma'(x) \leq \gamma'(y)$. Using definition of derivative and $0 < \gamma'(x)$, we obtain

$$0 < \frac{\gamma(x + \delta) - \gamma(x)}{\delta} \leq \frac{\gamma(y) - \gamma(y - \delta)}{\delta} \tag{21}$$

Multiplying the above inequality by $\delta$ and adding $\gamma(y - \delta) + \gamma(x)$ yields Eq. (20). Let $X$ denotes the set of elements of $\vec{x}$. We define $\Phi(\vec{x}) = \{x \in X | 0 < x < \beta\}$.

**Theorem 2.** *The size of the subset $\Phi$ for minimal vector is at most one.*
**Proof:** Consider $\vec{x}$ which satisfies Eqs. (17)–(18) and $|\Phi(\vec{x})| >$. In our proof, we define a method to transform $\vec{x}$ to $\vec{x'}$. Then we show the following properties for the transformation.

1. The value of $f_o$ for the transformed $\vec{x}$ is not greater than the original $\vec{x}$

$$f_o(\vec{x'}) \leq f_o(\vec{x}) \tag{22}$$

2. The transformation has convergence property. If we repeat the transformation we reach $\vec{x^*}$ where $\Phi(\vec{x^*}) \leq 1$ and we could not apply transformation anymore.

First the definition of transformation, because $|\Phi(\vec{x})| > 1$, we can find $x_i$ and $x_j$ from $\Phi(\vec{x})$ where $0 < x_i < \beta$ and $0 < x_j < \beta$. Without loss of generality, assume $0 < x_i \leq x_j < \beta$. We have two cases for $x_i + x_j$: (1) $x_i + x_j \geq \beta$, (2) $x_i + x_j < \beta$. In the first case, we define $\delta = \beta - y_i$ and then change only the value $x_i$ and $x_j$ in $\vec{x}$ to get $\vec{x'}$ as follows:

$$x_i' = x_i + \delta = \beta \tag{23}$$
$$x_j' = x_j - \delta \tag{24}$$

In the second case, we define $\delta = y_j$ and then change only the value $x_i$ and $x_j$ in $\vec{x}$ as follows

$$x_i' = x_i + x_j \tag{25}$$
$$x_j' = 0 \tag{26}$$

Note that $x_i' + x_j' = x_i + x_j$ and the constraint Eq. (17) is still satisfied by $\vec{x'}$.

After defining transformation, we prove the first property. Using Theorem 1 with defined $\delta$, we can conclude

$$\gamma(x_i + \delta) + \gamma(x_j + \delta) = \gamma(x_i') + \gamma(x_j') \leq \gamma(x_i) + \gamma(x_j) \tag{27}$$

Adding unchanged elements of $\vec{x}$ to both sides of above inequality, yields Eq. (22).
For the proof of the second property, consider Eqs. (23) and (26). These imply that at least one of $x_i$ or $x_j$ will not be in the subset $\Phi(\vec{x'})$. Then the size of subset $\Phi$ is decreased by the transformation as follows

$$|\Phi(\vec{x})| - |\Phi(\vec{x'})| = \begin{cases} 1 & \text{for } x_i \neq x_j \\ 2 & \text{for } x_i = x_j = \dfrac{\beta}{2} \end{cases} \tag{28}$$

Because $|\Phi(\vec{x})| > 0$ it will reach 1 or 0 eventually and more transformation is not possible. At this point, due to the first property Eq. (22) we reach a minimal vector.

Without loss of generality we assume that the elements of the minimal vector are ordered $x_1^* \geq x_2^* \geq \ldots \geq x_{|P|}^*$. The minimal (solution) of Eqs. (17) and (18) will be

$$x_i^* = \begin{cases} \beta & \text{for } 1 \leq i \leq n^* \\ \dfrac{\lambda}{\mu} - \beta n^* & \text{for } i = n^* + 1 \\ 0 & \text{for others} \end{cases} \qquad (29)$$

where $n^* = \left\lfloor \dfrac{1}{\beta\mu} \right\rfloor$ is the number of PMs which must be filled completely (up to $\beta$). The remaining load (if exists) must be dispatched to the next PM. For large scale data centers, we can neglect this PM and show the solution of Eqs. (10)–(13) as follows.

$$r_i^* = \begin{cases} \dfrac{1}{n^*} & \text{for } 1 \leq i \leq n^* \\ 0 & \text{for others} \end{cases} \qquad (30)$$

For implementation, we only need a random generator. When a new VM arrives, we draw $i$ form $[1, n^*]$ and sends this VM to PM $i$. We do not require any polling, therefore our implementation is simple and agile as we need it in a cloud data center.

## SIMULATION RESULTS

To validate the analytical solution, we have built a discrete event simulator of a CP data center using *MATLAB (2012)*. We have considered the system with two different sojourn time distribution: exponential and uniform. In both cases the mean sojourn time is fixed at $\mu = 10^1$. The mean inter arrival time of VMs was made variable from ($\lambda = 10^4$ to $10^8$) to give reasonable insight into the behavior and dimensioning of CP data centers. Regarding $\lambda$ and $\mu$, traffic intensity varies from $10^4$ to $10^8$ which represents the mean number of VMs in data center at steady state according to the Little formula. The number of active PMs according to $\left(n^* = \dfrac{\lambda}{\beta\mu}\right)$ depends indirectly on $\varepsilon$, e.g., for $\varepsilon = 10^{-5}$ ($\beta = 69$) it varies from 145 to 145,000 servers. The values chosen may be quite applicable to small- to large-sized CPs data centers that try to keep the utilization of their servers as high as possible while guarantee a minimum QoS for the users. It is noteworthy that no CP published information regarding average traffic intensity, number of servers or the percentage of reserved.

We generate confidence intervals (95%) for steady state measurement using independent replications with deletions method. First, we run 50 independent replications of each simulation, then we remove samples of transient phase and finally we calculate the sample mean. Figure 3 depicts the SLAV probability in data center at steady state. Simulation results follows analytical model perfectly for all $\left(\dfrac{\lambda}{\mu}\right)$ and $\beta$, values. This observation supports the validity of the analytical model findings. As can be seen, probability for exponential sojourn time is generally less than probability for uniform one. Note that the mean time is the same for both distribution, and this may relates to the variance. The variance of exponential is about 10, but the variance of uniform is about 30. As the number of active servers increases, the SLAV probability decreased steadily. This trend is due to the fact that with more active servers, an arriving VM has more places to be hosted and the chance of blocking and SLAV is lower.

Figure 4 shows the effect of sojourn time distribution on the convergence time. The data center with exponential sojourn time reaches to steady state sooner than a data center with uniform sojourn time. We only show the results for $\beta = 100$, $\lambda = 10^5$, $\mu = 10$ because the

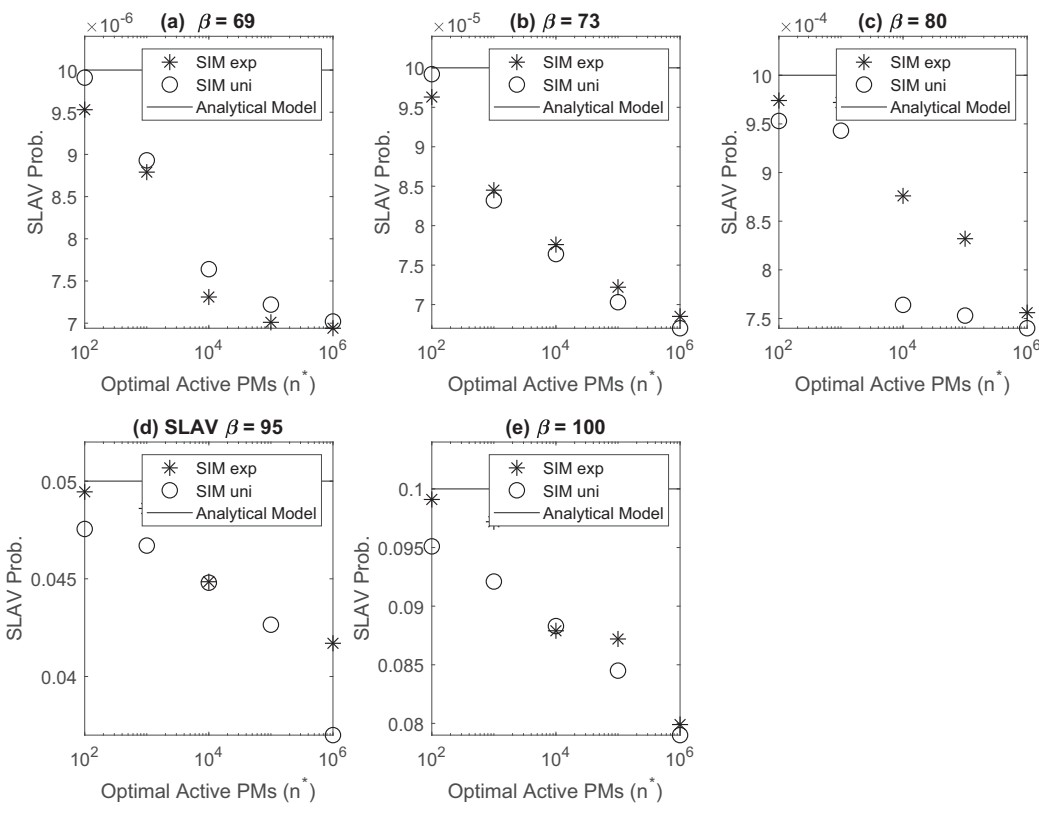

**Figure 3** SLAV probability in data center at steady state for various β (A) β = 69 (B) β = 73 (C) β = 80 (D) β = 95, (E) β = 100.

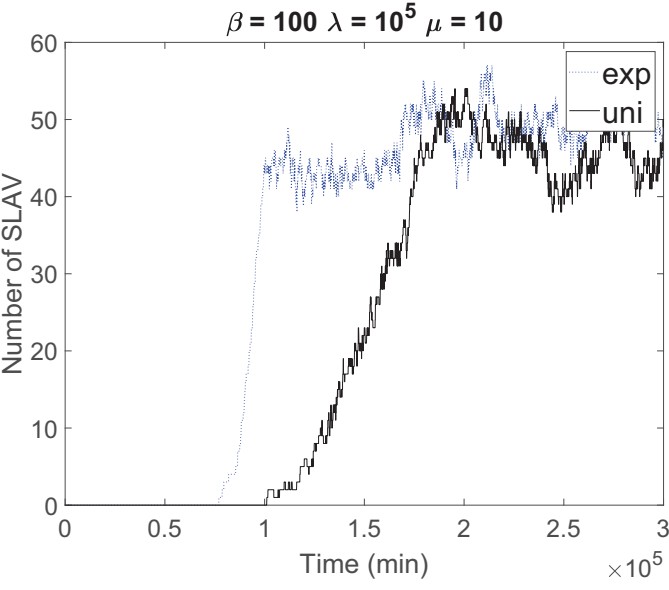

**Figure 4** Effect of sojourn time distribution on the convergence time.

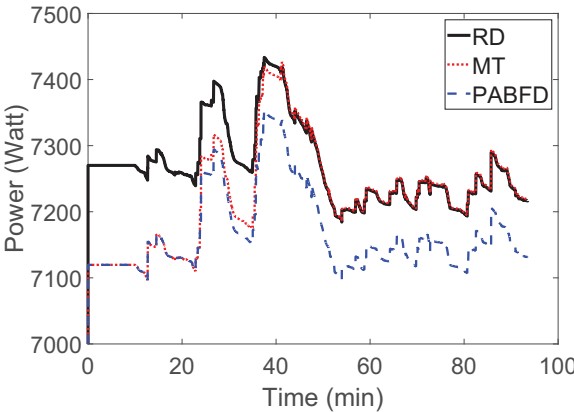

**Figure 5 Power consumption by a data center managed by random dispatcher and benchmark algorithms.**

discussion aims to highlight the influences of distribution on convergence time; for larger systems and lower probability, the discussion is the same.

In order to study the power consumption of our method in practice, we use Google trace (*Reiss, Wilkes & Hellerstein, 2011*) which consists of several concurrent VMs in a single ~12$K$ server farm where the resource demand of VMs is highly dynamic and needs quick decisions. Figure 5 compares the power consumption for our method with two benchmark algorithms in the literature: PABFD (*Beloglazov & Buyya, 2012*) and MT (*Domanal & Reddy, 2013*). The power consumption in our method is just about 2% more than PABFD and MT. This higher power consumption is acceptable, because the workload is highly dynamic, varying over time, and is driven by many short jobs that demand quick scheduling decisions and updates. PABFD and MT suffers from long decision process and overwhelming migrations. These shortcomings make both of them fruitless in real world cloud computing scenarios. The comparison gives us an idea about how far we are from idealistic benchmark algorithms in the literature.

## CONCLUSION

Effective resource management is a major challenge for the leading CPs (e.g., Google, Microsoft, Amazon). Performance evaluation of data centers is an important aspect of resource management which is of crucial interest for both CPs and cloud users. In this paper, we proposed an analytical model based on the $M/G/n/n$ system for performance evaluation of a cloud computing data center. Due to the nature of the cloud computing, we assumed general sojourn time for VMs as well as large number of PMs. These assumptions make our model acceptable in terms of scale and diversity. Through extensive numerical simulations, we showed that our analytical model closely alignes with simulation results. Our results also indicate that our method consumes a bit more power than idealistic benchmark in the literature.

In the future, we plan to extend our model for variable size VMs. Studying how "Power of Two Choice" can improve the result of randomized dispatcher will be another dimension of extension.

### Funding

The authors received no funding for this work.

### Competing Interests

The authors declare that they have no competing interests.

### Author Contributions

- Hadi Khani conceived and designed the experiments, performed the experiments, analyzed the data, contributed reagents/materials/analysis tools, prepared figures and/or tables, performed the computation work.
- Hamed Khanmirza analyzed the data, contributed reagents/materials/analysis tools, authored or reviewed drafts of the paper, approved the final draft, writing, Latex.

### Data Availability

The source code is available in the Supplemental File.

### Supplemental Information

Supplemental information for this article can be found online at http://dx.doi.org/10.7717/peerj-cs.211#supplemental-information.

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
