# Peer review of "Randomized routing of virtual machines in IaaS data centers"

_PeerJ Computer Science, doi:10.7717/peerj-cs.211_

## Round 0.1 · original submission · Major Revisions

This paper needs to give more background information on the problem that is being addressed and explain the simulation results better for the reviewers to be convinced that the proposed method is better.

Reviewer 1 ·

Basic reporting

Please see the general comments.

Experimental design

The experimental design is moderate.

Validity of the findings

Not convincing.

Additional comments

In this manuscript, the authors propose randomized method to assign the virtual machines to Physical Servers in IaaS data centers. The authors try to minimize the total power consumption under certain QoS constraints.

The reviewer has the following concerns:
1. The virtual machine allocation problem is not well defined. In the introduction, the authors should first introduce some background and then introduce the VM allocation problem.

2. The authors claim the the behaviour of a VM is unpredictable (which is the main motivation). It is better to explain what are the behaviours.

3. Towards the technical solution, the authors propose a randomized solution regardless of the current status of each Physical Server aiming at minimizing the total power consumption. The reviewer feels that the optimization problem is not well formulated, given the condition that all the VM will be randomly allocated.

4. There are many typos and grammar errors, such as ‘let to’ ‘be able of’.

Reviewer 2 ·

Basic reporting

The paper needs some good proofreading. Some examples here:
The cloud computing providers is running → are running
in data center -- in the data center
This let us to choose -- this let us choose
has been developed for initial placement of VMs -- for the initial placement
that PMs is fully utilized -- PMs are fully utilized
New request are either admitted or rejected → new request is
We suggest a randomized router after admission control phase. -- after the admission
Extensive analysis of huge data center -- huge data centers

Some figures lack units. (Figure 3, x-axis. Figure 5, y-axis)
There is a gap in Figure 5.

There is no explanation for many figures in the paper.

No background introduction. It would be better to introduce those compared methods (RR, TH, MT, BF).

Experimental design

no comment

Validity of the findings

The biggest problem for me is the simulation results. The authors compared their methods with several previous methods and the round robin method. However, from the Figure 5b, it seems that the proposed method works exactly the same as round robin, and worse than all previous work. Unless the authors can demonstrate that the proposed method performs better than those previous methods at least in some perspectives, I can’t find any advantages of the proposed one.

The proposed method is based on some very simple assumptions. For example, most VMs have approximately the same utilization; the power consumption is linear related to the utilization; feedback system is impossible in data centers. Those assumptions are not valid in real data centers.

Reviewer 3 ·

Basic reporting

The Abstract and Introduction are easy to read and follow.

However the rest of the sections are hard to follow.

One minor typo: Line 12: providers 'are' running

The references do not follow the standard template (e.g. line 180: Khani, H. (2016). Simulation code. Accessed: Nov 2016.)

Experimental design

'no comment'

Validity of the findings

It is not clear what is the challenge the authors are trying to address in the paper. The motivation of the paper is not clear.

Figure 3 is not clear. As beta decreases, the number of PMs increase and power consumption increases. So what is the author's recommendation from this figure? Same observation with Figure 5a.

The authors mention they use Google traces, but provide no further information of those traces.

---

## Round 0.2 · accepted · Accept

The revision has addressed the comments of the reviewer who had the most concerns.

Reviewer 2 ·

Basic reporting

None

Experimental design

None

Validity of the findings

None

Additional comments

The authors have fully addressed my concerns.